# *MAPK1* Is Regulated by LOC102188416/miR-143-3p Axis in Dairy Goat Mammary Epithelial Cells

**DOI:** 10.3390/genes13061013

**Published:** 2022-06-03

**Authors:** Yue Zhang, Jie Zhou, Shuang Liu, Zhibin Ji

**Affiliations:** 1School of Life Sciences, Technical University of Munich, 85354 Freising, Germany; zhang_yue_0624@163.com; 2Shandong Provincial Key Laboratory of Animal Biotechnology and Disease Control and Prevention, College of Animal Science and Veterinary Medicine, Shandong Agricultural University, Taian 271018, China; 18263820323@163.com (J.Z.); 17661213580@163.com (S.L.)

**Keywords:** miR-143-3p, lncRNA, *MAPK1*, mammary epithelial cell, goat, sequencing

## Abstract

MicroRNA-143-3p (miR-143-3p) is one of the miRNAs involved in the growth of goat mammary epithelial cells (GMECs). In this study, Illumina/Solexa sequencing was performed to establish the lncRNA database in Laoshan dairy goats. Using the lncRNA database, long noncoding RNAs (lncRNAs) regulated by miR-143-3p were screened. In total, 4899 lncRNAs were identified, with 173 lncRNAs being differentially expressed in all three replicates. The target genes of the differentially expressed lncRNAs were annotated in GO terms and KEGG pathways. Among the differentially expressed lncRNAs, lncRNA LOC102188416 was predicted to sponge miR-143-3p and share *MAPK1* as a common target gene with miR-143-3p, which was validated by dual luciferase reporter assay system and qRT-PCR. The miR-143-3p mimic significantly lowered the relative luciferase activity of psiCHECK2-LOC102188416 wildtype vector but not mutated vector, suggesting that lncRNA LOC102188416 might be a sponge of miR-143-3p, which was verified by the promotion role of lncRNA LOC102188416 siRNA (siR-LOC102188416) in the expression of miR-143-3p. It was shown that the expression of *MAPK1* was downregulated by either miR-143-3p mimic or siR-LOC102188416, indicating that miR-143-3p and lncRNA LOC102188416 had a coregulatory effect on *MAPK1* expression. The co-transfection of miR-143-3p inhibitor with siR-LOC102188416 reversed the decrease of *MAPK1* expression regulated by siR-LOC102188416 alone, strengthening the existence of lncRNA LOC102188416/miR-143-3p/*MAPK1* axis in GMECs of Laoshan dairy goats.

## 1. Introduction

MicroRNAs (miRNAs) are a group of endogenous noncoding RNA at a length of 18–25 nucleotides, which can lead to translational repression or mRNA degradation by binding to the 3′UTR of their target genes [1,2]. Studies have shown that miRNAs play significant roles in mammary gland development and lactation through the proliferation, apoptosis, and activity of mammary epithelial cells [3,4,5,6,7]. In our previous studies, miR-143-3p was found to be involved in the growth of goat mammary epithelial cells (GMECs) [8,9], and it has been reported that miR-143-3p promotes lipid synthesis in milk [10].

Long noncoding RNAs (lncRNAs) are noncoding transcripts longer than 200 nucleotides [11]. By competitively occupying the binding sites of miRNAs, lncRNAs can alter the expression of the target genes of miRNAs [12,13,14]. To explore the regulation of gene expression that relevant to GMEC growth, the interactions between miRNA and lncRNA need to be investigated. Mitogen-activated protein kinase 1 (*MAPK1*) is one of the members of *MAPK* family involved in proliferation, development, and differentiation [15,16]. Lu et al. revealed that *MAPK1* facilitates the synthesis of milk protein through STAT5 and mTOR pathways [17], and *MAPK1* can be targeted by miR-143-3p [18].

Previous studies have shown that *MAPK1* plays an important role in milk synthesis [17]. Investigating how *MAPK1* can be regulated would be helpful to understand the mechanism of milk synthesis and lactation, as well as the management of the genetic improvement and breeding of dairy goats. In this study, the lncRNAs regulated by miR-143-3p in GMEC were screened by Illumina/Solexa sequencing to explore the regulation between lncRNAs and miR-143-3p. Using the database, the potential miRNA sponges of miR-143-3p in differentially expressed lncRNAs (DE-lncRNAs) were screened, and the target genes of DE-lncRNAs were predicted. The lncRNA LOC102188416 sharing the same target gene, *MAPK1*, with miR-143-3p was selected. The regulation of *MAPK1* by lncRNA LOC102188416 and miR-143-3p, as well as the mutual regulation between LOC102188416 and miR-143-3p, was investigated.

## 2. Materials and Methods

### 2.1. Animals and Ethical Statement

Five Laoshan dairy goats from the Qingdao Aote original breeding farm were used. The goats were 4 years old, healthy, and under the same feeding and management. The mammary gland tissue around 30 g was cut with a scalpel from the five Laoshan dairy goats after general anesthesia, washed with phosphate-buffered saline, and stored in cold phosphate-buffered saline with penicillin/streptomycin for cell culture. The wound was sewn, and a regular sterilization was applied until the goats were recovered. All procedures conformed to the Institutional Animal Care and Use Committee of the Shandong Agricultural University (No. 2013005).

### 2.2. Cell Culture

The fresh mammary gland tissue was cut into pieces of approximately 1 mm^3^ and washed with PBS again before seeding into the plate. The goat mammary epithelial cells (GMECs) started to grow around the tissue pieces, and GMECs were purified via differential digestion. GMECs were cultured in DMEM/F12 medium (Gibco, Melbourne, VIC, Australia) with 10% fetal bovine serum (Gibco, Melbourne, VIC, Australia) and incubated in a humid environment with 5% CO_2_ at 37 °C. The transfection was performed by Lipofectamine 3000 (Invitrogen, Carlsbad, CA, USA) when GMECs reached 80% confluence. The synthesized small RNAs of SiR-LOC102188416, negative control for siRNA (siR-NC), miR-143-3p mimic, miR-143-3p inhibitor, and negative control (miR-NC) were purchased from RiboBio (Guangzhou, China).

### 2.3. RNA Isolation, Library Construction, Sequencing, and Data Analysis

Total RNA of GMECs was isolated using a MicroElute Total RNA kit (Omega Bio-tek, Norcross, GA, USA) according to the manufacturer’s instruction; the concentration of the RNA was measured using a nucleic acid spectrophotometer (DeNovix, Wilmington, DE, USA), while the quality and integrity of total RNA were detected by agarose gel electrophoresis and an Agilent 2100 Bioanalyzer (Agilent, Santa Clara, CA, USA), respectively. The NEB Next Ultra Directional RNA LibraryPrep Kit for Illumina (NEB, Ipswich, MA, USA) was applied to construct the library of lncRNA. Specifically, rRNA was removed from 3 μg of total RNA by Ribo-Zero GoldKits (Epicentre, Charlotte, NC, USA), and then fragmentation buffer was added to make RNA to short segments. The first strand of cDNA was synthesized by random hexamers, and the second strand of cDNA was acquired by dNTPs, RNase H, and DNA Polymerase I. The product of cDNA was purified with a QiaQuick PCR Purification Kit (Qiagen, Hilden, NRW, Germany), and, after end repair, base A addition, and sequencing adaptor addition, agarose gel electrophoresis was applied to collect the target size fragments. The cDNA strands were digested with Uracil-N-glycosylase, PCR amplification was performed, and the target size fragments were collected by agarose gel electrophoresis for library sequencing (Illumina/Solexa, San Diego, CA, USA).

The reads with low quality, adaptor contamination, rRNA mapping, or N proportion greater than 5% were removed from the raw sequencing data to obtain high-quality data. The high-quality data were mapped to the reference genome (https://www.ncbi.nlm.nih.gov/genome/?term=goat, accessed on 17 November 2017), and the expression abundance of lncRNAs was analyzed. The GO enrichment (http://geneontology.org/, accessed on 17 November 2017) and KEGG enrichment (https://www.genome.jp/kegg/, accessed on 17 November 2017) were performed for the predicted target genes of lncRNAs.

### 2.4. Real-Time Fluorescence Quantitative PCR

The cDNA for mRNA qPCR was synthesized using the PrimeScript RT reagent Kit with a gDNA Eraser (Takara, Japan) and the cDNA for miRNA qPCR was prepared using the Mir-X miRNA First-Strand Synthesis Kit (Takara, Japan). TB Green Premi Ex Taq II (Tli RNAseH Plus) (Takara, Japan) was applied to perform qPCR. The sequences of the primers are listed in Table 1.

### 2.5. Vector Construction and Dual Luciferase Reporter Assay

The sequence of lncRNA LOC102188416 containing the targeting site of miR-143-3p was amplified using goat DNA as template with primers 5′–agctttgtttaaacTTGTCCCAGCTCTACCCT–3′ (forward 1) and 5′–ataagaatgcggccgCCAAAGCAGCAAAGTTCCA–3′ (reverse 1), before inserting into the psiCHECK2 vector between Not I and Pme I restriction sites as the wildtype lncRNA LOC102188416 vector, named wt-LOC. Using the plasmid of the WT lncRNA LOC102188416 vector as the template, the targeting site was mutated by overlap PCR with primers 5′–CTGTCTCTAAAAGTGCAGCGTCATTGTGCT–3′ (forward 2) and 5′–CGCTGCACTTTTAGAGACAGCAGCTTCAGA–3′ (reverse 2). The mutated sequence was inserted into psiCHECK2 vector as the mutated LOC102188416 vector, named mut-LOC.

The wt-LOC and mut-LOC were co-transfected with miR-143-3p mimic or miR-NC, respectively, into GMECs in 24-well plates. The GMECs were lysed by a passive lysis buffer 48 h post transfection, and 20 μL of lysate was mixed into 100 μL of LAR II to measure firefly luciferase activity; then, 100 μL of Stop & Glo was added to measure *Renilla* luciferase activity. The ratio of *Renilla* luciferase activity to firefly luciferase activity was calculated to evaluate the relative luciferase activity.

### 2.6. Statistics

The experiments were performed three times independently. The data are shown as the average ± standard error (M ± SE). Student’s *t-*test and one-way ANOVA (SPSS 22.0; SPSS Inc., Chicago, IL, USA) were applied to evaluate the significance level (* *p* < 0.05, ** *p* < 0.01).

## 3. Results

### 3.1. Screening of Differentially Expressed LncRNAs Induced by miR-143-3p

To investigate the regulation between miR-143-3p and lncRNAs, miR-143-3p mimic and miR-NC were transfected into goat mammary epithelial cells (GMECs). The total RNA was extracted 48 h post transfection, and the expression abundance of lncRNAs was detected by Illumina/Solexa sequencing in three independent replicate experiments; 79,150,248 and 83,667,943 raw reads were obtained for the miR-143-3p and miR-NC groups, respectively. The raw reads were filtered, and 97.45% clean reads were acquired for further analysis. The coding ability of the transcripts was evaluated by CNCI (coding–noncoding index) [19], CPC (coding potential calculator) [20], Pfam (protein family) [21], and CPAT (coding potential assessment tool) [22], while the noncoding transcripts recognized by all four tools were identified as lncRNAs. In total, 4899 noncoding transcripts were identified by one of the four tools, of which 3828 lncRNAs were predicted by all four tools (Figure 1A; Appendix A). According to the database, 349 differentially expressed lncRNAs (DE-lncRNAs) were found between the miR-143-3p group and miR-NC group, including 75 upregulated lncRNAs and 274 downregulated lncRNAs (Figure 1B,C; Appendix A).

### 3.2. GO and KEGG Function Analysis of DE-LncRNAs

To estimate the function of the 349 DE-lncRNAs, the cis- and trans-target genes of DE-lncRNAs were predicted, and 9526 presumed target genes were obtained, including 767 cis-genes and 8759 trans-genes. In the GO enrichment analysis, 3156 genes were annotated in 2823 GO terms, and 174 GO terms were significantly enriched, including 72 biological processes, 63 cellular components, and 39 molecular functions, with the highest annotations for cellular process (GO:0009987), cell (GO:0005623), and binding (GO:0005488) (Figure 2A–C), while the most significantly enriched GO terms were metabolic process (GO:0008152), cell (GO:0005623), and binding (GO:0005488) (Figure 2D–F; Appendix A).

The KEGG enrichments showed that the predicted target genes of DE-lncRNAs were involved in 271 KEGG pathways, including PI3K/Akt signaling pathway (ko04151), MAPK signaling pathway (ko04010), mTOR signaling pathway (ko04150), and fatty-acid biosynthesis (ko00061)., while the significantly enriched pathways were peroxisome (ko04146), DNA replication (ko03030), cell cycle (ko04110), and aminoacyl-tRNA biosynthesis (ko00970) (Figure 3; Appendix A).

### 3.3. The Prediction of Target Genes of LncRNA LOC102188416/miR-143-3p

To find the DE-lncRNAs that were most likely to interact with miR-143-3p, DE-lncRNAs in three independent replicates were screened separately, and the 173 DE-lncRNAs in the intersection were selected (Figure 1C), where 70 of the 83 known lncRNAs were downregulated. Six of these downregulated lncRNAs were found to have binding sites to miR-143-3p: lncRNA LOC108633922, lncRNA LOC102185144, lncRNA LOC106503175, lncRNA LOC108637969, lncRNA LOC102188416, and lncRNA LOC108636992. The predicted target genes of the six lncRNAs were compared to predicted target genes of miR-143-3p, and it was found that lncRNA LOC102188416 and miR-143-3p shared a critical target gene, *MAPK1*. Therefore, lncRNA LOC102188416 was selected to explore its interaction with miR-143-3p.

### 3.4. The Construction of LncRNA LOC102188416/miR-143-3p/MAPK1 Axis

The sequence of lncRNA LOC102188416 containing the binding site of miR-143-3p was inserted into psiCHECK2 vector between the Not I and Pme I restriction sites, and the binding site was mutated as a control for the dual luciferase reporter assay (Figure 4A). The wildtype vector or the mutated vector was co-transfected into GMECs in vitro with miR-143-3p mimic or miR-NC, respectively, and the luciferase activities were measured using the lysate of GMECs harvested 48 h post transfection. The results showed that the miR-143-3p mimic significantly lowered the relative luciferase activity of the psiCHECK2-LOC102188416 wildtype vector but not the mutated vector (Figure 4B).

LncRNA LOC102188416 siRNA (siR-LOC102188416) and siR-NC were transfected into GMECs to detect the knockdown efficiency of siR-LOC102188416 (Figure 5A). Figure 5B shows that the expression of miR-143-3p was increased when lncRNA LOC102188416 was knocked down.

Figure 6 shows that the expression of *MAPK1* was decreased by siR-LOC102188416 (Figure 6A) and miR-143-3p (Figure 6B). The expression of *MAPK1* was decreased by siR-LOC102188416, while miR-143-3p inhibitor co-transfection with siR-LOC102188416 could eliminate the decrease in *MAPK1* expression induced by siR-LOC102188416 (Figure 6C).

## 4. Discussion

In the present study, miR-143-3p or miR-NC was transfected into GMECs of Laoshan dairy goat, and the lncRNA libraries were constructed to screen the differentially expressed lncRNAs (DE-lncRNAs) induced by miR-143-3p. LncRNA LOC102188416 was one of the DE-lncRNAs predicted to interact with miR-143-3p, both having *MAPK1* as a target. The interaction between lncRNA LOC102188416 and miR-143-3p, as well as their regulation of the expression of *MAPK1*, was explored.

It was revealed that the target sites of miRNAs can be competitively combined by endogenous competitive RNAs, such as circular RNAs [23], lncRNAs [24,25], and pseudogenes [26]; therefore, the activity of miRNAs can be modulated. LncRNA LOC102188416 was one of the lncRNAs downregulated by miR-143-3p, and it shared the same target gene, *MAPK1*, with miR-143-3p. It has been reported that *MAPK1* can upregulate milk protein synthesis [17], which is essential for milk production. Interestingly, *MAPK1* was also identified as a target gene of miR-940 [27], miR-362 [28], miR-585-3p [29], miR-378 [30], and miR-145 [31], and these miRNAs inhibit the proliferation of tumor or cancer cells through targeting *MAPK1*, which indicates that *MAPK1* could be targeted by multiple miRNAs to regulate cell proliferation.

Many previous studies have shown that miR-143-3p inhibits the progression of different kinds of cancers, such as breast cancer [32,33,34], ovarian cancer [35], cervical cancer [36], lung cancer [37], gastric cancer [38], and colorectal cancer [39]. In our previous study, it was shown that miR-143-3p suppresses the proliferation and facilitates the apoptosis of GMECs in vitro, which is consistent with the research in cancer cells [9]. However, the specific regulatory mechanism is not clear. LncRNAs, as an miRNA sponge, can adsorb miRNAs to regulate the expression of target genes and play a vital role in a variety of biological processes [36,37,38]. LncRNA LOC102188416 was identified as one of the lncRNAs downregulated by miR-143-3p, and it shared the same target gene, *MAPK1*, with miR-143-3p. *MAPK1* plays an important role in milk synthesis [17]. Investigating how *MAPK1* is regulated would be helpful to understand the mechanism of milk synthesis and lactation, as well as the management of the genetic improvement and breeding of dairy goats.

To investigate the regulatory mechanism of miR-143-3p, lncRNAs, which can absorb miRNAs to regulate the expression of target genes [36,37,38], regulated by miR-143-3p were studied. In this study, the lncRNA profiles were analyzed in GMECs cultured in vitro after transfection with miR-143-3p and its negative control, which could provide a reference for the study of lncRNAs in GMECs and their interaction with miR-143-3p. The lncRNA LOC102188416/miR-143-3p/*MAPK1* axis revealed that miR-143-3p is likely to be involved in mammary gland development and lactation by regulating lncRNAs and mRNAs, which may lay a theoretical foundation for the genetic breeding of dairy goats. The role of miR-143-3p and *MAPK1* in both healthy mammary epithelial cells and tumor or cancer cells could also be a consideration in further studies.

## 5. Conclusions

LncRNA profiles of Laoshan dairy goat mammary epithelial cells (GMECs) were analyzed between miR-143-3p and miR-NC groups. Among the lncRNAs identified, LncRNA LOC102188416 was significantly downregulated by miR-143-3p, and it could be a potential sponge of miR-143-3p. *MAPK1*, the target gene of miR-143-3p [18], was also predicted as a target gene of lncRNA LOC102188416. This research studied the interaction among lncRNA LOC102188416, miR-143-3p, and *MAPK1*, and the results revealed that lncRNA LOC102188416 acted as a sponge of miR-143-3p to regulate the expression of their mutual target gene *MAPK1* in GMECs.

## Figures and Tables

**Figure 1 genes-13-01013-f001:**
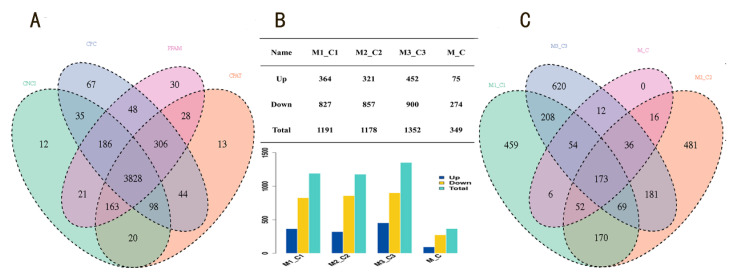
The identification of lncRNAs in different libraries of GMECs. (**A**) Number of identified lncRNAs by four tools. (**B**) The number of DE-lncRNAs between different groups. (**C**) Venn diagram of DE-lncRNAs between different groups. M1_C1, M2_C2, and M3_C3 represent the three independent replicates of miR-143-3p mimic vs. miR-NC. M_C represents the integrated analysis of M1_C1, M2_C2, and M3_C3.

**Figure 2 genes-13-01013-f002:**
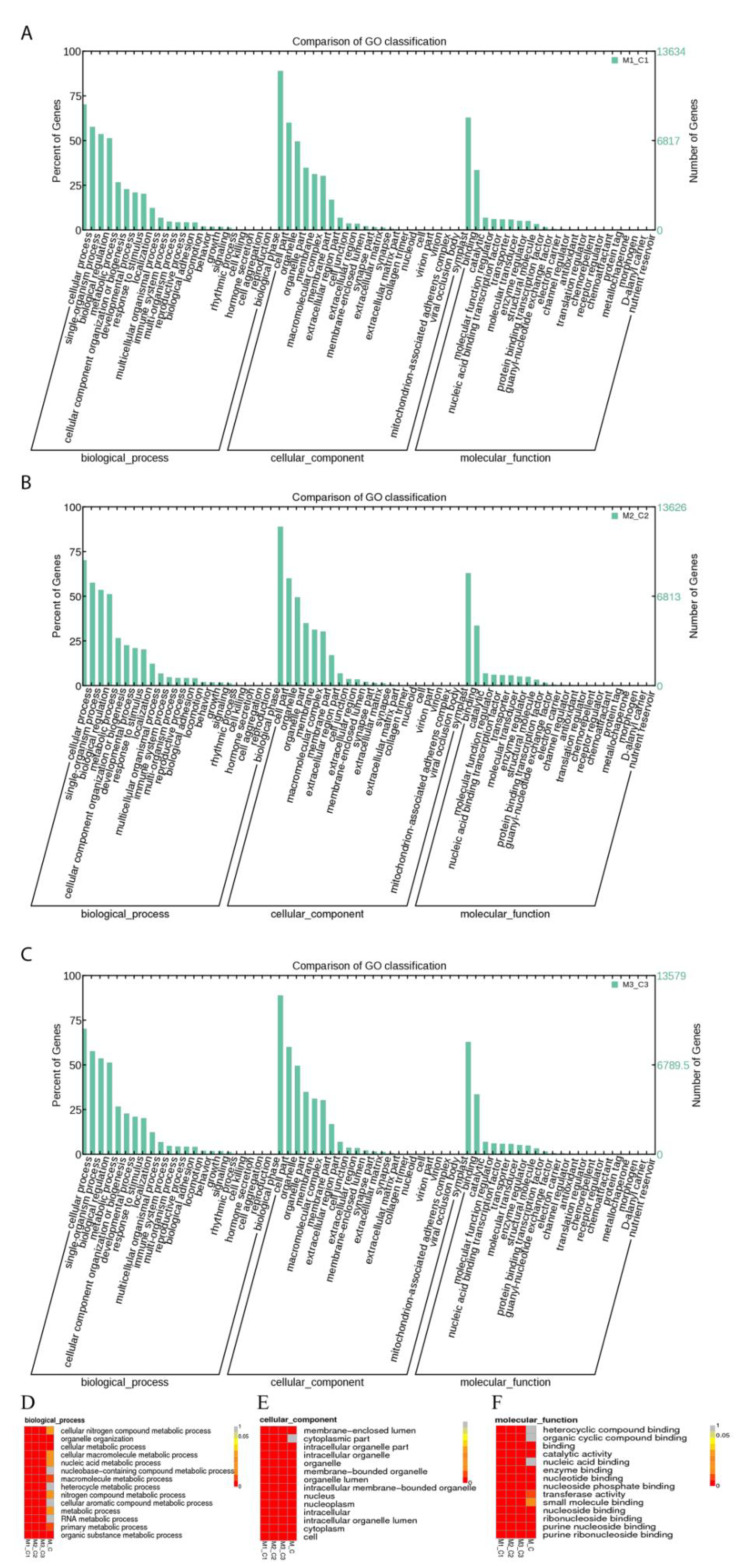
The enrichment analysis of predicted target genes of DE-lncRNAs. (**A**–**C**) The annotation classification in M1_C1 group (**A**), M2_C2 group (**B**), and M3_C3 group (**C**). (**D**–**F**) Significant enriched terms in biology processes (**D**), cellular components (**E**), and molecular functions (**F**).

**Figure 3 genes-13-01013-f003:**
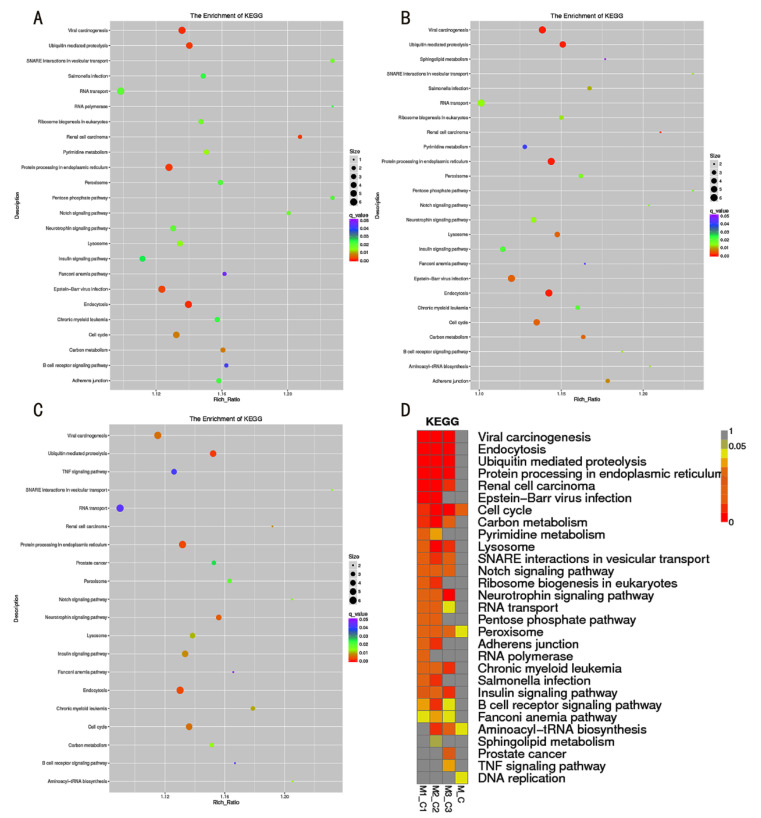
The KEGG enrichment results of predicted target genes of DE-lncRNAs. (**A**–**C**) The significantly enriched KEGG pathways in M1_C1 group (**A**), M2_C2 group (**B**), and M3_C3 group (**C**). (**D**) The heatmap of significantly enriched KEGG pathways in all groups.

**Figure 4 genes-13-01013-f004:**
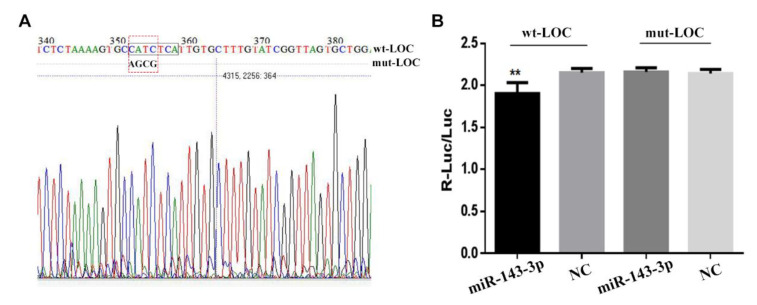
Sponge between miR-143-3p and lncRNA LOC102188416. (**A**) The construction of dual luciferase reporter vectors of lncRNA LOC102188416. (**B**) The luciferase activities of psiCHECK2-LOC102188416 vectors regulated by miR-143-3p. ** *p* < 0.01.

**Figure 5 genes-13-01013-f005:**
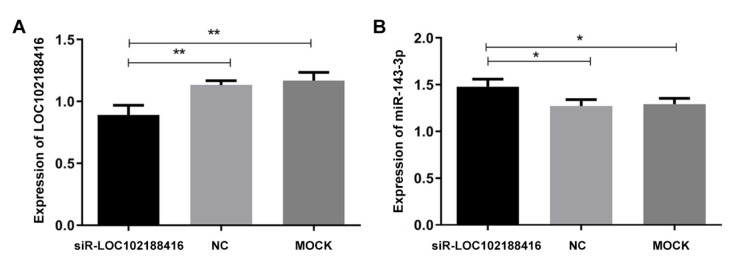
The regulation of miR-143-3p by lncRNA LOC102188416. (**A**) The efficiency validation of siR-LOC102188416. (**B**) The expression of miR-143-3p regulated by siR-LOC102188416. * *p* < 0.05; ** *p* < 0.01.

**Figure 6 genes-13-01013-f006:**
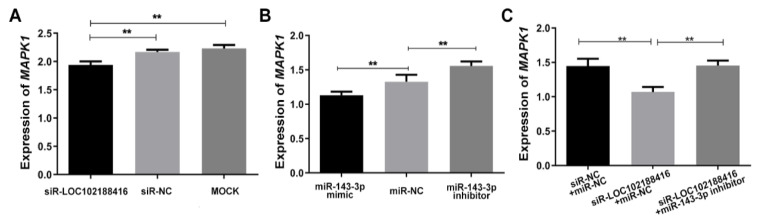
The regulation of *MAPK1* by lncRNA LOC102188416 and miR-143-3p. (**A**–**C**) *MAPK1* expression regulated by siR-LOC102188416 (**A**), miR-143-3p (**B**), and miR-143-3p inhibitor (**C**). ** *p* < 0.01.

**Table 1 genes-13-01013-t001:** The sequence of primers in qPCR.

Primer	Sequences (5′-3′)
lncRNA LOC102188416 forward	TGGGTTGAGGCACACTGGTCACC
lncRNA LOC102188416 reverse	CTCGCTTCGGCAGCACA
*MAPK1* forward	ACTGCCAGAGGACGCTGAGAG
*MAPK1* reverse	ATGTGGTCGTTGCTGAGGTGTTG
*GAPDH* forward	CACCCTCAAGATTGTCAGC
*GAPDH* reverse	CAGTGGTCATAAGTCCCTCC
miR-143-3p forward	TGAGATGAAGCACTGTAGCTCG
* U6 * forward	CTCGCTTCGGCAGCACA
* U6 * reverse	AACGCTTCACGAATTTGCGT

## Data Availability

All data in the manuscript are available through the responsible corresponding author.

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
