# Peer review of "MAPK1 Is Regulated by LOC102188416/miR-143-3p Axis in Dairy Goat Mammary Epithelial Cells"

_genes, 2022, doi:10.3390/genes13061013_

Round 1
Reviewer 1 Report
This is an interesting manuscript; however, significant alterations are required as suggest by the following comments.
Title: I suggest changing the title, at least move “Laoshan dairy goat” at the end.
Abstract: The text should be simplified and many terms e.g., “miRNAs” “siR” need to be defined at first use. Provide sample size. Gene names should be italic throughout the manuscript.
Introduction: There should be comprehensive introduction about the sample goat breed and already known genes and their roles.
Line 27: “Lactation” is a very broad term and cannot be regulated exclusively by “non-coding RNAs”. Therefore, authors should be very specific with reference to their experiments.
Lines 49-55: Change this section to present aims and hypothesis of this study.
Lines 138-139: Define “CNCI, CPC, PFAM, and CPAT” and provide references.
Figures: It is not possible to interpret most of the figures with the present size and quality. Moreover, Figure 2 description should be condensed and remove the repeating text “The annotation classification in” and “Significant enriched terms in”. Similarly, remove the repeating text from Figure 3.
Citations: There are two different citation styles, need to fix according to the journal guidelines. Many citations are missing in the references list.
Lines 229-233: Should move these sentences to Introduction > “Previous studies have shown that MAPK1 plays an important role in milk synthesis [14]. Investigating how MAPK1 can be regulated would be helpful to understand the mechanism of milk synthesis and lactation, and also the management of the genetic improvement and breeding of dairy goats.”
Lines 243-249: Surprisingly, this paragraph is copied/pasted of the lines 49-55.
Author Response
Dear Reviewer,
Thank you for your help to improve our manuscript.
We have modified the manuscript according to your constructive suggestions. If there is anything need to be improved, all comments are welcomed.
Best regards,
Yue Zhang & Zhi Bin Ji

Reviewer 2 Report
The article investigate the " MAPK1 is regulated by LOC102188416/miR-143-3p axis in 2 Laoshan dairy goat mammary epithelial cells"
The abstract needs to be rewritten in more understandable way , the information are their but the language should be improved along with simple and understandable flow information and to be linked with phenotypic info,.
Introduction
Lines from 30-31 paragraph should be improved (necessities not valid may be necessary , English editing is required)
Line 31 again increasing studies here is not suitable chose another word may be many studies …….
Line 34 the authors wrote " In our previous studies, miR-143-3p was found involved in the growth 34 of goat mammary epithelial cells (GMECs) [11, 12],, please rewrite ,,,
Line 46 reference number is missing
Lines 49- 55 , the first paragraph should go to M& M and please stat the objectives of the study in understandable way
Overall the introduction needs to language editing and more info should be added regarding the previous knowledge on the current investigation
Methodology
Please identify the abbreviation before using it eg. GMEC and PBS and for the others
Language editing is required
Results should be illustrated in an easy flow manner
Discussion needs more
No conclusion is found
The font size is not the same in all parts of the manuscript
English editing is required
Author Response
Dear Reviewer,
Thank you for your help to improve our manuscript.
We have modified the manuscript according to your constructive suggestions. If there is anything that needs to be improved, all comments are welcomed.
Best regards,
Yue Zhang & Zhi Bin Ji

Round 2
Reviewer 1 Report
Thank you for making the suggested changes. It could have been better to provide details of each change made according to the list of suggestions. Moreover, following changes are required to make it a quality article.
- Change figures to improve quality and readability of the text.
- Check spellings and grammar thoroughly.
- Conclusions should be around the title but not a simple rephrase of the title. It requires a few sentences to link your results with significant insights that should briefly support the title.
Author Response
Dear Reviewer,
Thank you for your help with the manuscript. We have checked the text overall and rewrote the Conclusion part. The resolution of Figure 2 has been improved to a better view. Hope it would be more reader friendly.
Best,
Yue